# Effectiveness of the Online “Dialogue Circles” Nursing Intervention to Increase Positive Mental Health and Reduce the Burden of Caregivers of Patients with Complex Chronic Conditions. Randomized Clinical Trial

**DOI:** 10.3390/ijerph20010644

**Published:** 2022-12-30

**Authors:** Jose Manuel Tinoco-Camarena, Montserrat Puig-Llobet, María Teresa Lluch-Canut, Juan Roldan-Merino, Mari Carmen Moreno-Arroyo, Antonio Moreno-Poyato, Judith Balaguer-Sancho, Zaida Agüera, Maria Aurelia Sánchez-Ortega, Miguel Ángel Hidalgo-Blanco

**Affiliations:** 1Center of Cornellà Specialists, Consorci Sanitari Integral, 08940 Barcelona, Spain; 2Department of Public Health, Mental Health and Maternal-Child Nursing, School of Nursing, University of Barcelona, Health Sciences Campus Bellvitge, L’Hospitalet de Llobregat, 08907 Barcelona, Spain; 3Department of Mental Health, Campus Docent Sant Joan de Déu School of Nursing, University of Barcelona, Sant Boi de Llobregat, 08830 Barcelona, Spain; 4Department of Nursing, Fundamental and Medical-Surgical, School of Nursing, Faculty of Medicine and Health Sciences, University of Barcelona, 08907 Barcelona, Spain; 5Department of Nursing Research Group (GRIN), Bellvitge Biomedical Research Institute (IDIBELL), L’Hospitalet de Llobregat, 08908 Barcelona, Spain; 6Hospital de la Santa Creu I Sant Pau, 08041 Barcelona, Spain; 7Psychoneurobiology of Eating and Addictive Behaviors Group, Neurosciences Programme, Bellvitge Biomedical Research Institute (IDIBELL), L’Hospitalet de Llobregat, 08908 Barcelona, Spain; 8CIBER Fisiopatología de la Obesidad y Nutrición (CIBERObn), Instituto Salud Carlos III, 28015 Madrid, Spain; 9University School of Nursing and Occupational Therapy of Terrassa (EUIT), Universitat Autònoma de Barcelona, 08221 Terrassa, Spain

**Keywords:** chronic disease, nursing intervention, clinical trial, dialogue circle, positive mental health, caregiver overload

## Abstract

The personal demands involved in caring for a chronically ill person can lead to emotional and physical exhaustion in caregivers. The aim of this study was to evaluate the effectiveness of an online nursing intervention called “dialogue circles” designed to reduce caregiver overload and enhance positive mental health (PMH) in family caregivers. We used a pre-post design. The sample consisted of 86 family caregivers of patients with complex chronic conditions, randomly assigned to the intervention group (n = 43) or the control group (n = 43). All participants completed the Zarit scale and the Positive Mental Health Questionnaire 15 days before starting the intervention and 30 days after its completion. Comparison of the post-test changes revealed statistically significant differences between the two groups in PMH and overload, with the intervention group showing greater positive changes in all dimensions of PMH after the intervention and lower scores on overload. In conclusion, the results suggest that incorporating dialogue circles as an online nursing intervention in the caregivers of patients with complex chronic conditions can enhance PMH and decrease caregiver overload, especially in settings where face-to-face encounters are not possible.

## 1. Introduction

Chronic diseases are becoming increasingly prevalent, as is the coexistence of several of such diseases in the same person [1]. These diseases often reduce functional capacity and lead to a high risk of cognitive decline, making affected individuals dependent for activities of daily living [2,3].

Patients with complex needs can be defined as those who are vulnerable, frail, with multiple comorbidities, difficult-to-control symptoms, a high risk of exacerbations, and taking multiple medications requiring integrated care to improve their quality of life. Equally, these patients require multiple healthcare resources and strong support from their families and a multidisciplinary healthcare team to ameliorate the suffering caused by the disease process [4].

The main resource in the care of dependent persons is the family [5]. Family carers often provide home care, with no specific training for this activity [6]. Continuous provision of this task and the lack of social and professional care often negatively affect principal carers [7] and increase their psychosocial burden [8,9,10]. Both caregivers and dependent persons may experience negative physical and psychological effects [11,12].

There is consensus-based evidence that persons carrying out the role of informal caregiving for a prolonged period perform multiple and complex tasks. The lack of training for the management of these tasks often provokes demoralization and stress [13,14], with negative effects such as depression, anxiety, sleep disturbance, and fatigue [15,16]. These difficulties form part of a process that constantly tests caregivers’ physical limits and mental health, often leading to overload and/or burnout. This, in turn, can negatively affect the relationship between caregiver and patient [16,17,18]. Moreover, recent studies also report that caregiver overload can reduce emotional wellbeing and positive mental health (PMH). Because of the time devoted to caregiving, these individuals often neglect their own needs, losing contact with their families and friends, which in turn leads to a pessimistic view of the future [19].

Several studies evaluating the role of the carers of persons with chronic diseases have concluded that the interventions needed should focus on empowering caregivers to perform their role, while maintaining optimal self-care [20,21,22], as well as on enhancing their wellbeing and PMH [23].

Various systematic reviews and meta-analyses concur on the difficulty of assessing the results on the effectiveness of interventions aimed at informal caregivers. This could be due to several factors such as the diversity of methods employed and the distinct designs and interventions used, as well as focus on a specific disease [24,25,26,27,28,29].

In this regard, some studies have used different types of intervention in carers, such as those combining self-help programs with psycho-educational programs [30], those based on the use of information and communication technology (ICT) [31,32,33], and those encouraging physical activity [34]. Among all these studies, some authors believe that psychotherapeutic interventions produce a greater benefit than psychoeducational interventions [35] and that multi-component group programs are more efficient in reducing carer overload [36,37].

Several authors have made proposals that include the use of ICTs in interventions aimed at informal caregivers [38,39]. Of note, one study used ICTs to enhance PMH in informal caregivers through the support of virtual platforms: (a) “Cuidadores 2.0” [Carers 2.0] through self-care resources [40] and (b) a program to encourage PMH through an app called “Cuidadoras crónicos” [Chronic carers]. This smartphone app-based intervention offered a different activity every day for 4 weeks. After the end of the intervention, the 3-month follow-up results showed an improvement in PMH [41].

Various studies have used new technologies to carry out nursing interventions in distinct populations [42,43,44], with favorable results. 

A novel intervention, which has not yet been explored in the health setting, is the use of dialogue circles; this intervention aims to offer carers support, advice, protection, and accompaniment during the process of change, in which they adjust to a life with the limitations imposed by caring for a patient with complex and chronic needs and with a chronic disease in an advanced stage. These circles are essentially spaces in which carers can share their feelings with other persons with similar preoccupations and fears; this provides an opportunity to normalize some thoughts and concerns as carers realize they are shared by other people in a similar situation. In addition, members of the group share strategies and information they have found useful in their role as caregivers [45]. Consequently, this intervention could be useful to enhance PMH and reduce caregiver overload. 

We found no previous studies applying this intervention in caregivers, although a similar online intervention, called “virtual culture circles” was used in families in Brazil to offer a space to encourage health promotion in coping with COVID-19 [46].

Given the gaps in the literature, the main objective of this study was to evaluate the efficacy of the “dialogue circles” nursing intervention to decrease caregiving overload and increase the PMH of primary caregivers of patients with complex chronic needs and advanced chronic disease.

## 2. Materials and Methods

### 2.1. Study Design 

This study was designed as a randomized controlled trial with simple masking. The study was registered in ClinicalTrials (ID: NCT04993248). The CONSORT checklist was used.

### 2.2. Setting and Participant Selection 

The study was conducted in primary health centers in Catalonia (Spain) administered by the publicly funded health system.

The sample was composed of the family carers of patients identified as having chronic complex disease or a chronic disease at an advanced stage, and who were found to have overload on the short-form version of the Zarit Burden Interview (ZBI-7) [47]. This questionnaire, derived from the ZBI [48], contains seven items on overload.

Participants were identified by nurse case managers. These nurses identified, among patients assigned to their services portfolio, the family caregivers who could be included in the study during routine nurse consultations in the health center or home. 

The inclusion criteria were as follows: (a) age > 18 years, (b) being a caregiver for 6 months or more, (c) experiencing caregiver overload according to the ZBI-7, (d) having internet access, and (e) not being involved in any type of individual or group psychological intervention during the performance of the intervention. Exclusion criteria were (a) being an occasional caregiver, (b) not having computer skills, and/or (c) not having the resources/media to connect to the online sessions. 

A total of 100 persons were invited to participate, of which 14 were excluded, mostly due to their own refusal (n = 10) or due to their inability to complete the intervention because of the health status of the person they cared for (n = 4). 

### 2.3. Randomization and Masking

Participants who met the inclusion criteria were contacted by telephone or in person during the medical visit to inform them of the study. Those who agreed to participate were randomly assigned to the control group (CG) or intervention group (IG) using the OxMaR software [49]. After assigning them to each group, they were given by hand or by e-mail a document containing the study information, self-reported questionnaires, and the informed consent document to be signed. Figure 1 shows a flow diagram of the distribution of the participants, following the CONSORT statement. The caregivers were unaware of their group assignment, which was performed using the single-blind technique.

### 2.4. Instruments

Data were collected through self-reported questionnaires sent through e-mail. 

An ad-hoc sheet was used to collect participants’ sociodemographic variables (age, sex, time since starting caregiving, educational level, marital status, number of children, employment situation, family relationship to the patient, whether living with the patient or not, and, if so, whether the patient received dependency benefits). 

The variable PMH was assessed through the Positive Mental Health Questionnaire (PMHQ; Lluch, 1999) [50]. This instrument contains 39 items with an unequal distribution between the 6 factors defining the construct: F1-Personal satisfaction (8 items), F2-Prosocial attitude (5 items), F3-Self-control (5 items), F4-Autonomy (5 items), F5-Problem solving and self-actualization (9 items), and F6-Interpersonal relationship skills (7 items). The items take the form of positive or negative statements that participants mark on a scale of 1 to 4, depending on the frequency with which they occur: always or almost always, fairly often, always or always, sometimes, never or not often. The questionnaire provides an overall PMH score (the sum of the scores of the elements), as well as specific scores for each factor. The overall PMH score ranges from 39 points (low PMH) to 156 points (high PMH). The minimum and maximum scores for each factor are as follows: 8–32 (factor F1), 5–20 (factors F2, F3, and F4), 9–36 (factor F5), and 7–28 (factor F6).

The questionnaire has been validated by several studies, obtaining a Cronbach’s alpha between 0.89 and 0.90 and a test-retest correlation of 0.85 [51,52]. Based on the validation of this questionnaire, a decalogue of recommendations to promote PMH was developed [53].

The ZBI-7, validated by Regueiro et al. (2007) [47], consists of 7 items related to overload and uses a Likert-like scale to obtain scores ranging from 7 to 35. In line with the literature, scores of 17 or more were considered to indicate intense overload. The questions focus on important areas such as caregiver health, psychological wellbeing, finances, social life, and the relationship between carer and patient [54].

The level of satisfaction in the IG was assessed using a satisfaction survey consisting of the following four questions with yes/no answers: (a) I felt understood, (b) I consider that the intervention has helped me to decrease overload, (c) I think all caregivers should undergo the intervention, and (d) I felt helped by listening to people in a similar situation. 

### 2.5. Intervention 

The intervention was performed between July 2021 and February 2022. Carers in the control group (CG) were provided with routine care and they were referred to social work to receive information on the help they could request. 

The IG (IC) underwent the dialogue circles intervention. This intervention aimed to increase PMH and reduce overload through the creation of safe spaces (circles) that allow participants to be protagonists and experience the trust necessary to voice problems, difficulties, and feelings surrounding their role as carers. 

The overall intervention consisted of conducting three dialogue circles per group. There were five groups: two composed of eight participants and three composed of nine participants and a facilitator, who was always the principal investigator of the study. Each circle lasted approximately 90 min, with a 15-day interval between each.

In the first circle, the facilitator explained the methodology and dynamics of the circles. The use of the item was explained: the item is a material object indicating whose turn it is to speak at that time and that the others are to pay attention. The object is passed around the circle by hand among the participants, providing each with an opportunity to share their thoughts and feelings if they wish. 

Receiving the object is an invitation to share with the group. The object helps to ensure that each participant in the circle has the opportunity to collaborate in their own time and way without being interrupted while they speak. Participants share what they want or remain silent during their turn, passing the object to the person next to them, from one participant to another, until the circle is closed. None of the circles has an established script and the themes to be discussed arise spontaneously among the participants. The facilitator intervenes if a participant monopolizes the circle. 

After this detailed explanation by the facilitator, the first circle starts with each participant introducing themselves, one after the other, applying the above-mentioned method and dynamics. The two remaining circles follow the same methodology and dynamics as the first. 

The circles were carried out online, through a video call using the Zoom platform. Consequently, the item was the raised hand icon.

Caregivers’ PMH and overload were measured in each group, 15 days before the start of the intervention and 30 days after its end, with measurements being performed at the same time in both groups. The IG was also administered a satisfaction survey on the intervention. 

### 2.6. Statistical Analysis 

The R.4.1.2 program was used for the statistical analysis. The baseline characteristics and outcomes (cross-sectionally and their change) of the IG and CG were compared using the chi-square test for categorical variables and the Student’s t-test, Fisher’s exact test, or Mann–Whitney U-test for quantitative variables. PMH was analyzed globally and by factors with direct scoring.

A repeated measures analysis of variance was performed to study the variation in the PMH and Zarit scales between the two study groups. An analysis of covariance with the independent variables age, educational level, and marital status was also performed.

The descriptive variables used were absolute and relative frequency for categorical variables, and median and interquartile range for continuous variables.

To evaluate the effect of the intervention on the change in pre-post intervention scores on the Zarit and PMH scales, regression models were fitted to explain the change in scores based on the intervention, age (with quadratic relationship), sex, baseline score, and the 2-sided interactions between the exposure group and the variables age, sex, and baseline score. A variable selection process based on Akaike information criterion was applied to obtain the final models.

## 3. Results

### 3.1. Sociodemographic Data

Overall, no differences were observed between the two groups at the baseline, except for age, education level, and marital status, with the IG being older and more frequently single and having a higher educational level than the CG (Table 1). For numerical variables, the median and interquartile range (Q1 and Q3, which contain 50% of the sample) are described, and for categorical variables, the absolute and relative frequency of each category. The *p*-value of the test of equal distribution of the variable between the control and intervention groups is presented: Wilcoxon if the variable is numerical, Fisher’s exact test for categorical variables.

### 3.2. Levels of Overload and Positive Mental Health at Baseline

At the baseline assessment of the scales (Table 2), the IG scored significantly higher on caregiver overload (assessed with the Zarit scale). There were no statistically significant differences in PMH in the overall PMHQ score between the two groups (Table 2). 

After categorization of the factors of the PMH, significant differences were found only in F3-self-control and F6-interpersonal relationship skills. 

### 3.3. Post-Intervention Levels of Overload and Positive Mental Health 

In the post-intervention assessment (Table 3), statistically significant differences were found between the CG and the IG, with the IG scoring higher in all factors of the PMHQ and overall and scoring lower on the ZBI-7 than the CG. 

### 3.4. Post-Intervention Changes 

Table 4 shows changes in the baseline and post-intervention scores. ZBI-7 scores significantly worsened by a median of 2 points in the CG and significantly improved by a median of 5 points in the IG. Overall PMH worsened in the CG but improved by a median of 18 points in the IG. Unlike the CG, the IG showed improvement in all factors at follow-up. This difference between the two groups was statistically significant (*p* < 0.001) (Table 4).

## 4. Discussion

Informal caregivers often have high levels of psychological distress and dissatisfaction in terms of meeting their own needs. Consequently, they require support and constant guidance by nurses [55], who can offer interventions that help to prevent complications in caregivers and reduce the risk of overload or ameliorate its effects [56]. Bearing in mind this premise, we performed this study to analyze the impact of the online “dialogue circles” nursing intervention on caregiver overload and PMH. As described, the main result revealed that this intervention produced positive changes in the dimensions of PMH and reduced the general level of overload reported by the family caregivers themselves.

The sociodemographic profile of the main caregivers in this study was in line with previous studies reporting that the typical profile of these individuals has the following characteristics: mainly middle-aged women, a first-degree relative of the patient (daughter or wife), with low to secondary level education, living with the patient, and with an average of two children [57,58,59,60,61,62,63]. In terms of the employment situation of the study population, most caregivers in this study had a paid job. This result is similar to that reported in another study [63], but contrasts with those of other authors [25,64] indicating that carers do not have paid work. Those results may be due to the large amount of time that carers spend in the care of older adults, limiting the time available to perform paid work. In this study, we did not examine whether the main caregivers received some form of financial aid to carry out their task (from other family members or some form of benefit), which could explain these discrepancies.

The results on the changes produced after the intervention show enhanced PMH and lower perceived overload in the IG. These results agree with those of another type of intervention in carers previously reported in the literature. A study by Ferré-Grau et al. (2021) [65] found that designing and implementing an ICT-based nursing intervention reduced overload and increased PMH in informal caregivers, with an increase in the global score of the PMH scale in the IG. The improvement in the PMH scores in the IG, both overall and in its different categories, was highly favorable and encouraging, since previous research had shown that PMH was correlated with self-care and, therefore, directly affected caregivers’ health and indirectly the quality of care provided to the patient [51,52]. Therefore, the present study demonstrates that the dialogue circles nursing intervention may benefit not only carers but also patients. 

To our knowledge, no previous studies have analyzed the efficacy of a dialogue circles nursing intervention in informal caregivers. However, there have been previous reports of a similar technique in other study populations. For example, a study by Silva et al. (2019) [66] implemented the culture circles created by Freire (1987) [67] in teachers, while Souza et al. (2021) [46] used them in a nursing intervention in the families of patients with COVID-19, showing that these tools encourage the ability to reflect on participants’ behavior by allowing them to be the protagonists of their own health/disease story, thus enhancing their quality of life. Although dialogue circles have not been previously used as a nursing intervention, Souza et al. (2021) [46] showed that culture circles can help to reduce the overload perceived by the main carers of patients with complex chronic needs and with advanced-stage disease, by offering a space in which they can share their experiences and feelings, which in turn encourages empathy as participants feel identified with others while listening to them describe their own similar situations. Therefore, this type of intervention may be beneficial for other carers. 

Satisfaction with the dialogue circles nursing intervention was high, with all participants reporting they were satisfied, since they indicated that they had felt understood and that the experience had helped them to reduce their feeling of overload. Moreover, they felt that the intervention should be extended to all family caregivers. These results are similar to those reported by Lleixà-Fortuño et al. (2015) [40], who designed a web 2.0 website for the carers of patients with chronic problems and who found that user satisfaction was higher than 93% and that a similar percentage of carers would recommend the website to other caregivers. Likewise, Ferré-Grau et al. (2021) [65] found that satisfaction with their app-based intervention was high among users, who stated that they would recommend it to other carers. A large proportion of the users agreed that increasing the duration of the program would be beneficial.

### Limitations and Strengths

This study should be considered in the context of several limitations. First, all outcome variables (i.e., perceived overload, PMH, and satisfaction) were assessed using self-reported measures. Although all the questionnaires used have been validated and have good psychometric properties [6,26,50], we have to point out that self-reported measures may also have biases that can affect the results, such as social desirability bias. Secondly, the effectiveness of the intervention has not been assessed in terms of objective variables, such as the reduction in the number of hospital readmissions and the number of emergency visits of affected family members, as well as the improvement in the physical health or economic situation of caregivers. These limitations should be addressed in future studies, and longer-term follow-ups would be useful to assess the consistency of the reported improvements.

Despite these limitations, the study also presents several noteworthy strengths. To our knowledge, this is the first study exploring the effectiveness of the dialogue circles nursing intervention. It is a novel intervention delivered online and useful for reducing family caregiver overload and improving their PMH by creating a space for social support and an opportunity for group reflection.

## 5. Conclusions

The results of this study suggest that nursing intervention in online dialogue circles improves PMH and reduces the perception of caregiver overload. It offers satisfaction to caregivers, as it makes them visible and allows them to interact with other people who are going through the same situation. Likewise, the use of new technologies to carry out nursing interventions favors the participation of caregivers, avoiding displacements and saving them time.

Our study is, to our knowledge, the first to evaluate the nursing intervention of dialogue circles. It is a novel intervention—all the more so because it is conducted online—that appears to be effective in reducing family caregiver overload. The intervention may also help to improve their PMH by creating a space that provides social support and an opportunity for reflection in this group.

However, further studies are needed to determine whether this type of intervention, whether face-to-face or online, can contribute to improving the care of patients with chronic diseases and their families and thus respond to the current situation of social and health care.

## Figures and Tables

**Figure 1 ijerph-20-00644-f001:**
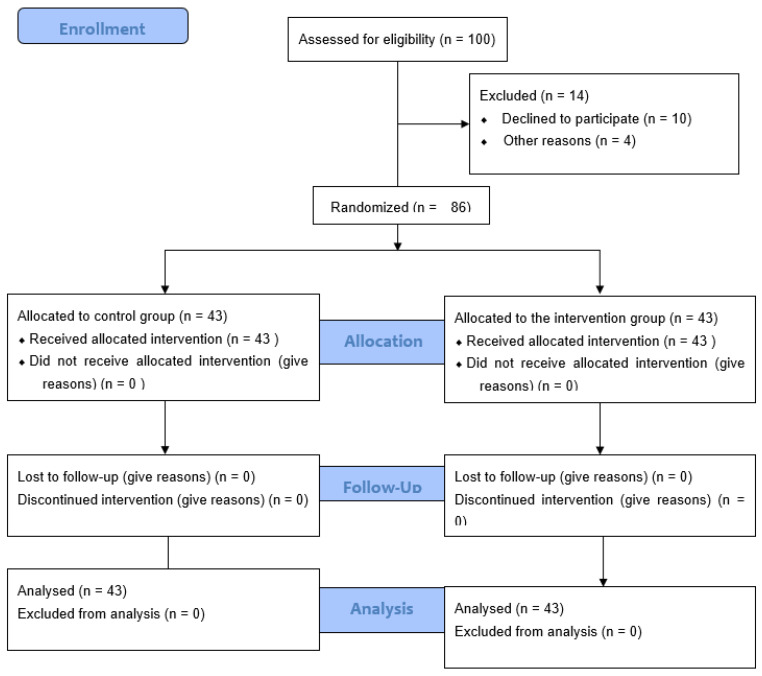
Flow diagram showing participants’ progress through the phases of the randomized two-arm parallel group clinical trial.

**Table 1 ijerph-20-00644-t001:** Baseline characteristics of the sample, overall and by group.

Items		Overall	Control	Intervention	*p*-Value
Age		56.00 [49.25, 63.00]	60.00 [54.50, 65.00]	53.00 [46.50, 59.00]	0.002
Caregiver months		26.50 [16.25, 48.00]	30.00 [18.00, 48.00]	24.00 [14.00, 48.00]	0.267
Gender					0.102
	Men	17 (19.77%)	12 (27.91%)	5 (11.63%)	
	Women	69 (80.23%)	31 (72.09%)	38 (88.37%)	
Educational level					<0.001
	No studies	1 (1.16%)	1 (2.33%)	0 (0.00%)	
	Primary education	21 (24.42%)	17 (39.53%)	4 (9.30%)	
	Intermediate level training cycle	14 (16.28%)	2 (4.65%)	12 (27.91%)	
	Baccalaureate or higher education	23 (26.74%)	11 (25.58%)	12 (27.91%)	
	University education	27 (31.40%)	12 (27.91%)	15 (34.88%)	
Marital status					0.030
	Single	12 (13.95%)	3 (6.98%)	9 (20.93%)	
	Domestic partner	6 (6.98%)	1 (2.33%)	5 (11.63%)	
	Married	56 (65.12%)	34 (79.07%)	22 (51.16%)	
	Divorced	12 (13.95%)	5 (11.63%)	7 (16.28%)	
Number of children					0.171
	0	23 (26.74%)	10 (23.26%)	13 (30.23%)	
	1	19 (22.09%)	7 (16.28%)	12 (27.91%)	
	2	38 (44.19%)	24 (55.81%)	14 (32.56%)	
	3	6 (6.98%)	2 (4.65%)	4 (9.30%)	
Employment status					0.244
	Unemployed	15 (17.44%)	8 (18.60%)	7 (16.28%)	
	Part-time work	5 (5.81%)	2 (4.65%)	3 (6.98%)	
	Employed	44 (51.16%)	18 (41.86%)	26 (60.47%)	
	Self-employed	6 (6.98%)	5 (11.63%)	1 (2.33%)	
	Retired	16 (18.60%)	10 (23.26%)	6 (13.95%)	
Family relationship					0.277
	Spouse	17 (19.77%)	11 (25.58%)	6 (13.95%)	
	Son/daugher	67 (77.91%)	32 (74.42%)	35 (81.40%)	
	Son-in-law/daugher-in-law	1 (1.16%)	0 (0.00%)	1 (2.33%)	
	Others	1 (1.16%)	0 (0.00%)	1 (2.33%)	
Living together					0.459
	No	22 (25.58%)	13 (30.23%)	9 (20.93%)	
	Yes	64 (74.42%)	30 (69.77%)	34 (79.07%)	
Receives financial benefits					1.000
	No	65 (75.58%)	33 (76.74%)	32 (74.42%)	
	Yes	21 (24.42%)	10 (23.26%)	11 (25.58%)	

**Table 2 ijerph-20-00644-t002:** Direct scores and by levels on baseline scales in the overall sample and in the control and intervention groups.

Items	Overall	Control	Intervention	*p*-Value
ZBI-7 Total score	24.00 [21.00, 27.00]	22.00 [20.00, 25.00]	26.00 [24.00, 28.00]	<0.001
Burden	86 (100.00%)	43 (100.00%)	43 (100.00%)	1.000
PMH total score	93.00 [89.25, 96.00]	93.00 [90.50, 96.50]	91.00 [89.00, 95.50]	0.467
F1: Personal satisfaction	15.00 [13.00, 18.00]	15.00 [13.00, 18.00]	16.00 [14.00, 19.00]	0.252
F2: Prosocial attitude	14.00 [13.00, 15.00]	14.00 [13.00, 14.50]	14.00 [13.00, 15.00]	0.446
F3: Self-control	13.00 [11.00, 14.00]	14.00 [12.00, 14.50]	11.00 [10.00, 13.00]	<0.001
F4: Autonomy	8.50 [7.00, 11.00]	8.00 [7.00, 10.00]	9.00 [7.00, 11.00]	0.238
F5: Problem solving and self-actualization	26.50 [23.00, 30.00]	28.00 [23.50, 31.00]	26.00 [22.00, 28.50]	0.029
F6: Interpersonal relationship skills	15.00 [13.00, 16.00]	14.00 [13.00, 15.00]	15.00 [14.00, 17.00]	0.005

PMH: positive mental health; ZBI-7: 7-item Zarit Caregiver Burden.

**Table 3 ijerph-20-00644-t003:** Follow-up scores for the overall sample and for the control and intervention group.

Items	Overall	Control	Intervention	*p*-Value
ZBI-7 Total score	23.00 [20.00, 26.00]	25.00 [22.00, 27.00]	20.00 [18.00, 24.00]	<0.001
Burden	80 (93.02%)	43 (100.00%)	37 (86.05%)	0.034
PMH total score	98.00 [89.25, 112.00]	89.00 [83.00, 93.00]	112.00 [105.50, 118.00]	<0.001
F1: Personal satisfaction	17.50 [14.00, 21.00]	14.00 [12.00, 18.00]	20.00 [17.00, 22.50]	<0.001
F2: Prosocial attitude	15.00 [13.00, 16.00]	13.00 [12.00, 14.00]	16.00 [16.00, 17.00]	<0.001
F3: Self-Control	13.00 [12.00, 15.00]	13.00 [11.00, 14.00]	15.00 [13.00, 16.00]	<0.001
F4: Autonomy	10.00 [8.00, 12.00]	8.00 [6.00, 10.00]	12.00 [9.00, 14.00]	<0.001
F5: Problem solving and self-actualization	28.50 [24.00, 32.00]	25.00 [19.50, 29.00]	32.00 [28.00, 33.50]	<0.001
F6: Interpersonal relationship skills	16.00 [14.00, 19.00]	14.00 [13.00, 15.00]	19.00 [18.00, 21.00]	<0.001

PMH: positive mental health; ZBI-7: 7-item Zarit Caregiver Burden.

**Table 4 ijerph-20-00644-t004:** Changes between baseline and post-intervention scores.

Items	Control	Intervention	*p*-Value
ZBI-7 total score	2.00 [1.00, 3.00]	−5.00 [−6.00, −4.00]	<0.001
PMH total score	−4.00 [−8.50, −1.50]	18.00 [14.00, 25.50]	<0.001
F1: Personal satisfaction	0.00 [−1.00, 0.50]	3.00 [2.00, 4.50]	<0.001
F2: Prosocial attitude	−1.00 [−1.00, 0.00]	2.00 [1.00, 3.00]	<0.001
F3: Self-control	−1.00 [−2.00, 0.00]	2.00 [1.50, 4.00]	<0.001
F4: Autonomy	0.00 [−0.50, 1.00]	2.00 [1.00, 3.00]	<0.001
F5: Problem solving and self-actualization	−2.00 [−4.50, 0.00]	5.00 [3.00, 8.00]	<0.001
F6: Interpersonal relationship skills	0.00 [−1.00, 1.00]	4.00 [3.00, 5.00]	<0.001

PMH: positive mental health; ZBI-7: 7-item Zarit Caregiver Burden.

## Data Availability

The data that support the findings of this study are available from the corresponding author [M.P.-L.], upon reasonable request.

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
