# Peer review of "Effectiveness of the Online “Dialogue Circles” Nursing Intervention to Increase Positive Mental Health and Reduce the Burden of Caregivers of Patients with Complex Chronic Conditions. Randomized Clinical Trial"

_ijerph, 2022, doi:10.3390/ijerph20010644_

Round 1

Reviewer 1 Report

This paper aimed to compare the effectiveness of the “Dialogue Circles” versus usual care on mental health and burden among the caregivers caring the patients with complex chronic conditions. A randomized pre-post design was used to compare the two groups: the intervention group (n=43) versus the control group (n=43). The primary results of this study showed the effect of Dialogue Circles was superior to usual care in improving mental health and decreasing caregiver overload. Overall, this manuscript was generally nice and might be interesting to the readers. However, the authors need to do some efforts in statistical analysis and results. I offer the following comments to enhance its overall clarity, quality and value to the readership and beyond.

Abstract

1.       “Effectiveness” was used in the title, while “the efficacy of an online” was used in the Abstract. Efficacy is not the same as effectiveness. The authors should clarify this point.

2.       The authors may consider “a randomized pre-post design” to describe the research design (line 38).

Materials and methods

3.       Line 152, would expect the reference for OxMaR software

4.       Line 138, the inclusion criteria (a) age < 18 years? Was it typo?

5.       Statistical analysis

a.       Please clarify the statistical methods were used to examine the effectiveness of the “Dialogue Circles” versus usual care on mental health and burden among the caregivers caring the patients with complex chronic conditions? The current version on the statistical method section only described the comparisons of sociodemographic and overload and mental health status at baseline if I interpreted correctly.

b.       Based on the results from Table 1, the participants in the IG were older and had a higher educational level than those in the CG. As age and educational level may influence the outcomes of interest, they should be accounted for the effects of covariates to increase the precision of the effect of interest.

c.        In addition, given treatment effectiveness might vary depending on baseline mental health and overload status, the baseline-by-treatment interactions were also included when examining the effect of interest

Results/tables

6.       There were no missing data across tables. Thus, table columns of missing numbers could be removed.

7.       Please defined the table columns and include its units if possible. For examine, Table 1, the control group, age of 60.00 was for the mean age. And what did [54.50, 65.00] stand for?

Discussion

8.       Please include the study limitations in the discussion section.

Author Response

We deeply appreciate each of the comments made as a reviewer. Below are the responses to each comment. 

 Abstract

“Effectiveness” was used in the title, while “the efficacy of an online” was used in the Abstract. Efficacy is not the same as effectiveness. The authors should clarify this point.

We thank the reviewer for the comment. As suggested by the reviewer, the term "efficacy" has been changed to "effectiveness" throughout the manuscript. Indeed, the term "effectiveness" is more appropriate because it assesses the objectives/results ratio under real conditions. (“Efficacy” is instead under ideal conditions).

  1. The authors may consider “a randomized pre-post design” to describe the research design (line 38).

The “pre-post randomized design” has been detailed to describe the research design in the abstract (line 31).

Materials and methods

  1. Line 152, would expect the reference for OxMaR software

As the reviewer suggested, the reference for OxMaR has been added. (OxMaR [49]) 

O’callaghan, C.A. OxMaR: Open-Source Free Software for Online Minimization and Randomization for Clinical Trials., doi:10.1371/journal.pone.0110761.

4. Line 138, the inclusion criteria (a) age < 18 years? Was it typo?

Thanks to the reviewer for pointing out this typo, it has been corrected (see now ">18 years").

Statistical analysis

  1. Please clarify the statistical methods were used to examine the effectiveness of the “Dialogue Circles” versus usual care on mental health and burden among the caregivers caring the patients with complex chronic conditions? The current version on the statistical method section only described the comparisons of sociodemographic and overload and mental health status at baseline if I interpreted correctly.
  2. Based on the results from Table 1, the participants in the IG were older and had a higher educational level than those in the CG. As age and educational level may influence the outcomes of interest, they should be accounted for the effects of covariates to increase the precision of the effect of interest.
  3. In addition, given treatment effectiveness might vary depending on baseline mental health and overload status, the baseline-by-treatment interactions were also included when examining the effect of interest

We appreciate the reviewer's comment.  The R.4.1.2 program was used for the statistical analysis. The baseline characteristics and outcomes (cross-sectionally and their change) of the IG and CG were compared using the chi-square test for categorical variables and the Student t-test, Fisher Exact test or Mann-Whitney U test for quantitative variables. PMH was analyzed globally and by factors with direct scoring.

A repeated measures analysis of variance was performed to study the variation in the PMH and Zarit scales between the two study groups. An analysis of covariance with the independent variable age, educational level and marital status was also performed.

The descriptive variables used were absolute and relative frequency for categorical variables, and median and interquartile range for continuous variables.

To evaluate the effect of the intervention on the change in pre-post intervention scores on the Zarit and PMH scales, regression models were fitted to explain the change in scores based on the intervention, age (with quadratic relationship), sex, baseline score, and the 2-sided interactions between the exposure group and the variables: age, sex, and baseline score. A variable selection process based on Akaike Information Criteria was applied to obtain the final models.

Results/tables

There were no missing data across tables. Thus, table columns of missing numbers could be removed.

We appreciate this constructive comment. Therefore, the columns of sample size (n) have been removed through all the Tables in the manuscript.

Please defined the table columns and include its units if possible. For examine, Table 1, the control group, age of 60.00 was for the mean age. And what did [54.50, 65.00] stand for?

As suggested by the reviewer, all values in the Tables have been defined both in the Statistical Analysis section and in the Tables where the corresponding footnotes have been included to allow their interpretation.

For numerical variables, the median and interquartile range (Q1 and Q3, which contain 50% of the sample) are described, and for categorical variables, the absolute and relative frequency of each category. The p-value of the test of equal distribution of the variable between the control and intervention groups is presented: Wilcoxon if the variable is numerical, Fisher's exact test for categorical variables.

Discussion

  1. Please include the study limitations in the discussion section.

As the reviewer suggests, the following statement has been included in the “Limitations and strengths” section: 

4.1 . Limitations and strengths

This study should be considered in the context of several limitations. First, all outcome variables (i.e., perceived overload, PMH, satisfaction) were assessed using self-reported measures. Although all the questionnaires used have been validated and have good psychometric properties [6,26,50., we have to point out that self-reported measures may also have biases that can affect the results, such as social desirability bias. Secondly, the effectiveness of the intervention has not been assessed in terms of objective variables, such as the reduction in the number of hospital readmissions and the number of emergency visits of affected family members, as well as the improvement in the physical health or economic situation of caregivers. These limitations should be addressed in future studies., longer-term follow-ups would be useful to assess the consistency of the reported improvements.

Despite these limitations, the study also presents several noteworthy strengths. To our knowledge, this is the first study exploring the effectiveness of the dialogue circle nursing intervention. It is a novel intervention delivered online and useful for reducing family caregiver overload and improving their PMH by creating a space for social support and an opportunity for group reflection.

Thank you again for your input and comments. Best regards. 

Reviewer 2 Report

The paper deals with a very important and very current topic about efficacy of an online nursing intervention called "dialogue circles" designed to reduce caregiver overload and enhance positive mental health in family caregivers.

Very well written paper.

The title reflects the study but is too long. The authors could use some synthesis skills to improve it.

Introduction very well framing the problematic and with connection to the aim of the study.

Chapter 2. Material and Methods explains the methodology used and with a good description of the data collection instruments.

Chapters 3. Results and 4. Discussion very objective and well developed in order to meet the objective of the study.

I missed a sub-chapter on the limitations and implications of the study.

The authors should refer to the care they took in the fulfillment of the ethical principles in the text of the paper and not only in the Institutional Review Board Statement. To further develop this component of the respect for ethical principles.

The Conclusions should be more developed, considering the objective of the study and the support given in the discussion.

A vast set of bibliographical sources, very adequate to the type of study and more than 50% up to date.

I congratulate the authors for the paper.

Author Response

We deeply appreciate each of the comments made as a reviewer. Below are the responses to each comment.

1 The title reflects the study but is too long. The authors could use some synthesis skills to improve it.

We fully agree with the reviewer's comment that the title is very long and we have tried to synthesize it, but in the end we have kept the title because despite being long, as the reviewer himself points out, it reflects the study carried out and we think that it can facilitate the reader's understanding of what the study contains.

2 I missed a sub-chapter on the limitations and implications of the study.

As suggested by the reviewer, a Limitations and Strengths section has been included in the manuscript.

4.1 . Limitations and strengths

This study should be considered in the context of several limitations. First, all outcome variables (i.e., perceived overload, PMH, satisfaction) were assessed using self-reported measures. Although all the questionnaires used have been validated and have good psychometric properties [6,26,50., we have to point out that self-reported measures may also have biases that can affect the results, such as social desirability bias. Secondly, the effectiveness of the intervention has not been assessed in terms of objective variables, such as the reduction in the number of hospital readmissions and the number of emergency visits of affected family members, as well as the improvement in the physical health or economic situation of caregivers. These limitations should be addressed in future studies., longer-term follow-ups would be useful to assess the consistency of the reported improvements.

Despite these limitations, the study also presents several noteworthy strengths. To our knowledge, this is the first study exploring the effectiveness of the dialogue circle nursing intervention. It is a novel intervention delivered online and useful for reducing family caregiver overload and improving their PMH by creating a space for social support and an opportunity for group reflection.

3 The authors should refer to the care they took in the fulfillment of the ethical principles in the text of the paper and not only in the Institutional Review Board Statement. To further develop this component of the respect for ethical principles

We understand the reviewer concern. Therefore, the following statement has been included in the manuscript: In accordance with the Declaration of Helsinki, the present study was approved by the Ethics Committee of our institution [Clinical Research Ethics Committee of the Instituto de Investigación en Atención Primaria Jordi Gol (reference 20/112-PCV)]. All the participants provided signed informed consent.

The Conclusions should be more developed, considering the objective of the study and the support given in the discussion

The conclusions have been expanded

The results of this study suggest that nursing intervention in online dialogue circles improves PMH and reduces the perception of caregiver overload. It offers satisfaction to caregivers, as it makes them visible and allows them to interact with other people who are going through the same situation. Likewise, the use of new technologies to carry out nursing interventions favors the participation of caregivers, avoiding displacements and saving them time.

Our study is, to our knowledge, the first to evaluate the nursing intervention of dialogue circles. It is a novel intervention, all the more so because it is conducted online that appears to be effective in reducing family caregiver overload. The intervention may also help to improve their PMH by creating a space that provides social support and an opportunity for reflection in this group.

However, further studies are needed to determine whether this type of intervention, whether face-to-face or online, can contribute to improving the care of patients with chronic diseases and their families and thus respond to the current situation of social and health care.

A vast set of bibliographical sources, very adequate to the type of study and more than 50% up to date. I congratulate the authors for the paper.

We deeply appreciate the reviewer's constructive comments.

Thank you again for your contributions and comments, which have been very appropriate and constructive. 
Best regards. 

Reviewer 3 Report

Review of a manuscript -Manuscript ID: ijerph-2103976

The topic of the work is interesting and encourages you to get acquainted with the work.

However, in the work I see several shortcomings:

It is not clear how people were recruited for the study. This non-obviousness may also result from the translation.

There were no criteria for the exclusion of participants from the studies.

In addition, the authors themselves write about the differences between the groups ("...except for age, education level and marital status, with IG being older and more often single and having a higher level of education than CG..."). And it is precisely such sociodemographic characteristics that may affect the results of the study. Please refer to this description.

Please correct the word "alllow" on line 243. It's more about the word "allow".

Reference: item 66 - appears to be incomplete. Please complete it or refer to this comment.

The work requires corrections before being allowed for publication.

Author Response

We deeply appreciate each of the comments made as a reviewer. Below are the responses to each comment. 

1 It is not clear how people were recruited for the study. This non-obviousness may also result from the translation.

We understand the reviewer concern. Therefore, the following statement has been included in the manuscript: Participants who met the inclusion criteria were contacted by telephone or in person during the medical visit to inform them of the study. Those who agreed to participate were randomly assigned to the control group (CG) or intervention group (IG) using the OxMaR software [49].  After assigning them to each group, they were given by hand or by e-mail a document containing the study information, self-reported questionnaires, and the informed consent document to be signed. Figure 1 shows a flow diagram of the distribution of the participants, following the CONSORT statement. The caregivers were unaware of their group assignment, which was performed using the single-blind technique.

2 There were no criteria for the exclusion of participants from the studies.

As suggested by the reviewer, the exclusion criteria were added to the manuscript as follows: Exclusion criteria were a) being an occasional caregiver, b) not having computer skills, and/or c) not having the resources/media to connect to the online sessions

3 In addition, the authors themselves write about the differences between the groups ("...except for age, education level and marital status, with IG being older and more often single and having a higher level of education than CG..."). And it is precisely such sociodemographic characteristics that may affect the results of the study. Please refer to this description.

To evaluate the effect of the intervention on the change in pre-post intervention scores on the Zarit and PMH scales, regression models were fitted to explain the change in scores based on the intervention, age (with quadratic relationship), sex, baseline score, and the 2-sided interactions between the exposure group and the variables: age, sex, and baseline score. A variable selection process based on Akaike Information Criteria was applied to obtain the final models.

4 Please correct the word "alllow" on line 243. It's more about the word "allow".

Done. Thanks to the reviewer for advising of the typo.

5 Reference: item 66 - appears to be incomplete. Please complete it or refer to this comment.

Reference 66 (now 67) which was incomplete has been completed

Thank you again for your contributions and comments, which have been very appropriate and constructive. 
Best regards.